# Chemiluminescent screening of specific hybridoma cells via a proximity-rolling circle activated enzymatic switch

Hang Ao[1], Weiwei Chen[1], Jie Wu [1✉], Wencheng Xiao[1] & Huangxian Ju [1✉]

The mass-production capability of hybridoma technology is bottlenecked by the routine screening procedure which is time-consuming and laborious as the requirement of clonal expansion. Here, we describe a 1-day chemiluminescent screening protocol for specific hybridoma cells on conventional 96-well plate via a proximity-rolling circle activated enzymatic switch (P-RCAES) strategy. The P-RCAES uses a pair of antigen-DNA probes to recognize secreted specific antibody and proximity-induce rolling circle amplification for mass-production of pyrophosphate to activate Cu(II) inhibited horseradish peroxidase and generate a strong chemiluminescent signal. The P-RCAES based homogeneous chemiluminescent assay can detect antibody down to 18 fM, and enables the screening of specific hybridoma cells secreting PCSK9 antibody at single-cell level without tedious cloning process. The proposed fast screening protocol has good expansibility without need of sophisticated instruments, and provides a screening method for greatly improving the efficiency of hybridoma technology.

[1] State Key Laboratory of Analytical Chemistry for Life Science, School of Chemistry and Chemical Engineering, Nanjing University, Nanjing 210023, China.
✉email: wujie@nju.edu.cn; hxju@nju.edu.cn

Monoclonal antibodies that have been proven to be the powerful therapeutic agents against various diseases can be produced by different methods such as hybridoma technology, phage display, ribosome display, mRNA display, and B-cell immortalization technology[1–3]. Among these methods, hybridoma technology is the most classic to produce high-quality monoclonal antibodies. However, the antibody production process is labour-intensive and time-consuming because the hybridoma technology includes aspects of animal immunization, cell fusion, hybridomas screening and subcloning, as well as antibody characterization. The key to improve the efficiency of this technology is to develop efficient screening methods for identifying and selecting specific antibody-secreting hybridoma cells, because the hybridoma cells generally obtained either by PEG- or electrofusion are heterogeneous cells including specific antibody-secreting cells, nonspecific antibody-secreting cells, or nonsecretors[2,3]. Although commonly used ELISA is mature and reliable, its limited sensitivity requires the clonal expansion for 1–2 weeks to obtain enough secreted antibody for detection[4–7]. Thus, this technology is unsatisfied for quick and early screening. Recently, several droplet-based microfluidic techniques have been developed for single-cell screening by encapsulating individual cells into droplets with picoliter-nanoliter volume, in which the secreted antibody can reach a concentration up to $\mu g\,mL^{-1}$ within a few hours for sandwich immunoassay[8–11]. Although these techniques exhibit the capability of high throughput and efficient cell screening or sorting, their popularization is hampered by the sophisticated and expensive equipment as well as the specialized operation. Hence, the development of simple, convenient, and sensitive antibody assay systems is still an enormous challenge in efficient screening of hybridoma cells.

Homogeneous analysis is charming for antibody detection since it can be carried out in solution without the need of separation, immobilization, and washing. Some molecular switch sensors, which integrate the "capture" and "detection" units in one nucleic acid[12,13] or protein molecule[14–16], have been developed as an ideal means to perform the homogeneous antibody detection. The detection units can be switched on via the binding-induced conformational change to directly express the target binding event. However, these sensors normally lack sufficient sensitivity for antibody detection. Another kind of popular homogeneous analytical methods for protein detection was proposed through proximity binding-induced DNA assembly[17,18]. They use a pair of aptamers or antibody-DNA probes to recognize target protein, which results in proximity hybridization to form a proximity-ligated sequence for further DNA assemblies, such as polymerase chain reaction[19,20], rolling circle amplification (RCA)[21,22], nuclease/ribozyme-mediated cycle amplification[23,24], catalytic hairpin assembly[25,26], and hybridization chain reaction[27]. These assemblies can greatly amplify the detectable chemiluminescent[23,27], electrochemical[25], or fluorescent[26,28–31] signals for highly selective and sensitive protein analysis. In view of the advantages of homogeneous chemiluminescent detection technology, such as low cost, high sensitivity, high throughput, and operation convenience in multiplexed assay of DNA, miRNA, or proteins[32,33], here we design a proximity-rolling circle activated enzymatic switch (P-RCAES) to develop an ultrasensitive chemiluminescent protocol for homogeneous antibody detection and quick screening of specific hybridoma cells.

RCA is a rapid and isothermal technique and is usually used to extend the primer oligonucleotide along a circular template up to hundreds of times for generating the amplified signal[34–36]. In this work, the newly designed P-RCAES uses pyrophosphate anions (PPi), the extensively produced by-product of RCA, to activate Cu(II) inhibited horseradish peroxidase (HRP-Cu$^{2+}$) for

generating a chemiluminescent signal. In P-RCAES, the specific antibodies secreted from the hybridoma cell have been recognized by a pair of antigen-DNA probes which induces the proximity hybridization for forming a proximity-ligated sequence complementary to block. Upon the hybridization of this sequence with block, the primer can be released from block-primer to trigger the RCA. As a proof-of-concept, this enzymatic switch amplified chemiluminescent assay (CLA) has been proposed for detecting the secreted proprotein convertase subtilisin/kexin type 9 antibody (PCSK9-Ab), an efficient antibody drug for cardiovascular disease[37,38]. The homogeneous CLA can detect PCSK9-Ab down to 18 fM with a wide quantitative range, thus it can be applied to screen the specific hybridoma cells at single cell level on common 96-well plate through detecting the secreted PCSK9-Ab. Moreover, the whole screening procedure of specific hybridoma cells can be completed within 1 day, providing a candidate for greatly improving the efficiency of hybridoma technology.

## Results and discussion

### Chemiluminescent screening of specific antibody-secreting hybridoma cells.
Aiming at developing an efficient screening system with simplicity, universality and low-cost, we proposed the chemiluminescent screening protocol for specific hybridoma cells on common 96-well plate (Fig. 1a). After heterogeneous hybridoma cells were distributed on 96-well plate to allow single cell in a well by limiting dilution, the plate was incubated for 1 day to secret the target antibody from specific hybridoma cells. The CLA of the antibody secreted in the supernatant of each hybridoma cell was then performed to screen specific antibody-secreting hybridoma cells. It was worth saying that no cell proliferation was observed in such an incubation period.

In the chemiluminescent screening protocol, the secreted antibody was detected by its recognition to a pair of antigen-DNA probes (Supplementary Table 1) to trigger the P-RCAES (Fig. 1b). In P-RCAES, a complex with a proximity-ligated oligonucleotide sequence was firstly formed by the antibody-induced proximity hybridization of two antigen-DNA probes (Supplementary Fig. 1). The formed sequence then hybridized with block to release the primer from block-primer (Supplementary Table 1). In the presence of padlock template, splint R ligase, phi 29 polymerase, and dNTPs, the RCA could be initiated by the released primer. Here, splint R ligase could ligate the padlock to the ring template, and phi 29 polymerase could extend the primer along the ring template with dNTPs to produce long oligonucleotide amplicon[34] as well as a huge mass of PPi, an excellent chelate agent for metal ions[39–41], which could take away Cu$^{2+}$ from HRP-Cu$^{2+}$ to switch on the enzymatic activity of HRP for catalysing the oxidation of luminol by $H_2O_2$. The produced strong chemiluminescent signal was thus used for the detection of secreted antibody and the identification of specific hybridoma cells.

### Feasibility of P-RCAES.
In this work, the enzymatic switch was designed based on the inhibition of metal ion ($M^{n+}$) to HRP activity and the recovery of HRP activity by removing $M^{n+}$ from HRP-$M^{n+}$ with PPi (Fig. 2a). In order to achieve highly sensitive enzymatic switch, both high inhibition ability of $M^{n+}$ to HRP and high recovery ability of PPi to HRP-$M^{n+}$ should be ensured. The chemiluminescent intensities of HRP ($I_0$), $M^{n+}$-inhibited HRP ($I_i$), and recovered HRP ($I_r$) in the mixture of luminol and $H_2O_2$ were observed (Supplementary Fig. 2), and the chemiluminescence decrease percentage (($I_0$-$I_i$)/$I_0$) and chemiluminescence increase ratio ($I_r/I_i$) were calculated to evaluate the inhibition ability of $M^{n+}$ to HRP activity and the recovery ability of PPi to HRP-$M^{n+}$, respectively (Fig. 2b). Many ions including $Al^{3+}$, $Fe^{3+}$, $Cu^{2+}$, and $Cu^+$ showed chemiluminescence decrease

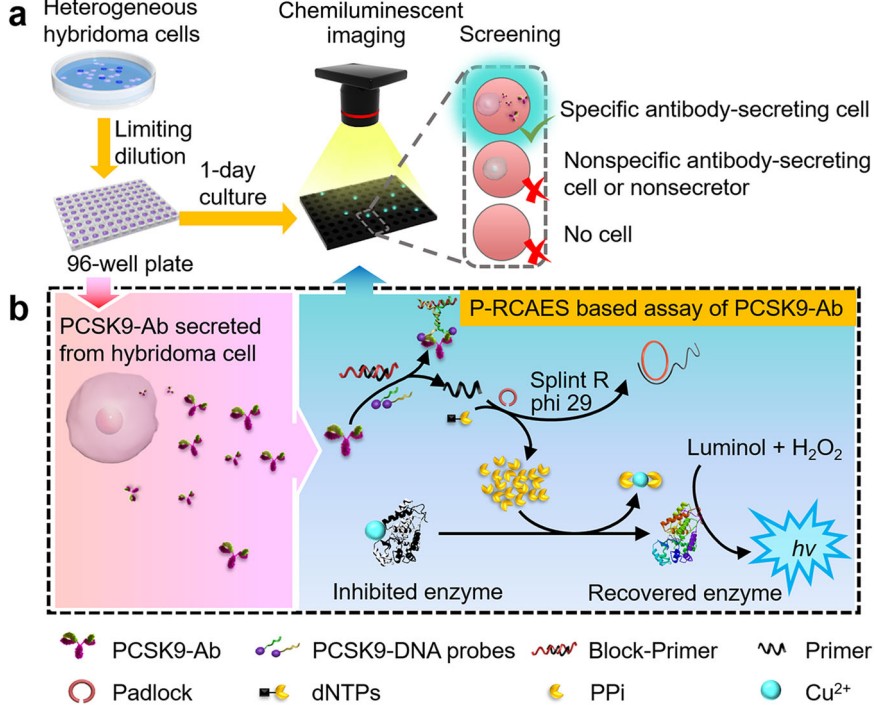

**Fig. 1 Illustration of the chemiluminescent screening of specific hybridoma cells with proximity-rolling circle activated enzymatic switch. a** Distribution of heterogeneous hybridoma cells on 96-well plate into single cell by limiting dilution to culture for 1 day, and then chemiluminescent assay of PCSK9-Ab in supernatants of the single hybridoma cell samples for screening of specific hybridoma cells. **b** chemiluminescent assay of secreted PCSK9-Ab via proximity-rolling circle activated enzymatic switch.

percentage higher than 92%, however, obvious chemiluminescence increase by PPi was only observed on $Cu^{2+}$ inhibited HRP (8 times increase), hence HRP-$Cu^{2+}$ was adopted to construct enzymatic switch. One thing to be noted, $Cu^+$ was a better HRP inhibitor than $Cu^{2+}$[42], it could inhibit HRP activity well even at a concentration 30 times lower than that of $Cu^{2+}$, however, the poor recovery of HRP activity from HRP-$Cu^+$ made it hard to design enzymatic switch.

Compared with the completely quenched chemiluminescent emission of HRP-$Cu^{2+}$, a slight decrease of the chemiluminescent response was observed after the addition of $Cu^{2+}$ into the mixture of HRP with luminol and $H_2O_2$ (Fig. 2c, column III), which could be attributed to the relatively slow conjugation of HRP-$Cu^{2+}$ and also excluded the possibility of $Cu^{2+}$ as a quencher of chemiluminescent emission. Moreover, the obvious recovery of chemiluminescent signal of HRP-$Cu^{2+}$ by PPi (Fig. 2c, column IV) and negligible effect of $CuPPi_2$ on the activity of HRP (Fig. 2c, column V) both indicated PPi had stronger coordination effect than HRP on $Cu^{2+}$ and the activation of HRP due to the removing of $Cu^{2+}$ from HRP-$Cu^{2+}$ via the chelate of $Cu^{2+}$ and PPi. Additionally, the conjugation of HRP-$Cu^{2+}$ did not change the chemiluminescent emission wavelength (Supplementary Fig. 3) and the absorption peak position of TMB-$H_2O_2$ system (Supplementary Fig. 4), indicating the same enzymatic mechanisms as those of HRP, which was further demonstrated by the similar decay rates (Supplementary Fig. 5).

Interestingly, the P-RCAES reaction mixture in the presence of PCSK9-Ab could also recover the $Cu^{2+}$ inhibited chemiluminescent response (Fig. 2d), indicating the generation of PPi due to the triggered RCA. Compared to the $Cu^{2+}$ inhibited chemiluminescent response (Fig. 2c), the slight increase of chemiluminescent response in the absence of PCSK9-Ab resulted from the presence of dNTPs, which could be considered as the background for PCSK9-Ab detection (Fig. 2d, column II). These results

demonstrated the feasibility of P-RCAES for detection of monoclonal antibody.

**Optimization for PPi-activated enzymatic switch**. To gain the best performance of PPi-activated enzymatic switch, the solution pH and concentrations of $H_2O_2$ and luminol were optimized. With the increasing pH of the mixture of HRP or HRP-$Cu^{2+}$ with luminol and $H_2O_2$, the chemiluminescent response increased, however, the maximum recovery efficiency of chemiluminescent response by PPi occurred at pH 7.2 (Supplementary Fig. 6a), which was close to the physiological pH for guaranteeing the activity of target antibody. Thus, following measurements were performed at pH 7.2. Similarly, the optimal concentrations of $H_2O_2$ and luminol were 1 mM and 0.2 mM respectively, at which the system showed the maximum chemiluminescence recovery efficiency (Supplementary Fig. 6b, c).

**Chemiluminescent response to PPi and primer via activation of enzymatic switch**. The performance of enzymatic switch was evaluated by examining the chemiluminescent response of HRP-$Cu^{2+}$ to PPi (Fig. 3a). After HRP-$Cu^{2+}$ was incubated with different amounts of PPi, the reaction mixtures were used to catalyze the chemiluminescent reaction of luminol and $H_2O_2$. Because the relative chemiluminescent intensity change ($\Delta I$, referring to the absolute signal subtracting the background) was entirely dependent on the target concentration, it was used to establish the linear detection curves. Here, $\Delta I$ increased logarithmically with the increasing concentration of PPi over a range of $10^{-9}$–$10^{-3}$ M (Fig. 3b, c). Such a concentration range of 6 orders of magnitude for PPi provided a good path for developing the analytical methods to detect PPi-related substances. As PPi was a by-product of RCA process, we anticipated the PPi-activated

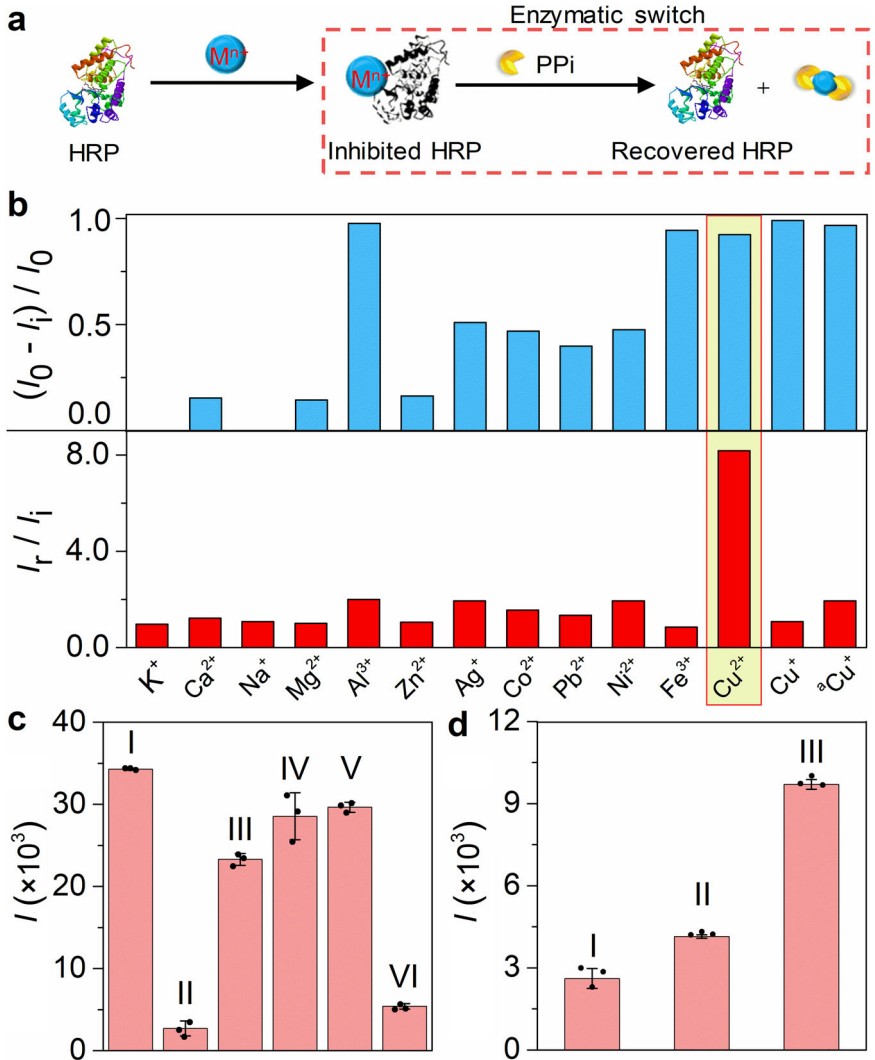

**Fig. 2 Design of the enzymatic switch and feasibility of proximity-rolling circle activated enzymatic switch. a** Schematic diagram of the pyrophosphate anions-activated enzymatic switch. **b** Comparison of enzymatic switches constructed with different metal ions. **c** Chemiluminescent intensity ($I$) of 10 mM pH 7.2 Tris-HCl containing 0.2 mM luminol, 1 mM $H_2O_2$ and (I) 0.2 μg mL$^{-1}$ HRP, (II) 0.2 μg mL$^{-1}$ HRP-Cu$^{2+}$, (III) 0.2 μg mL$^{-1}$ HRP and 0.12 mM Cu$^{2+}$ or (IV,V,VI) reaction mixture of 0.2 μg mL$^{-1}$ HRP-Cu$^{2+}$ and 2 mM pyrophosphate anions (IV), 0.2 μg mL$^{-1}$ HRP and 0.12 mM CuPPi$_2$ (V) and 0.2 μg mL$^{-1}$ HRP-Cu$^{2+}$ and 2 mM dNTPs (VI) after 5-min incubation. **d** Chemiluminescent intensity of 10 mM pH 7.2 Tris-HCl (50 μL) containing 0.2 mM luminol, 1 mM $H_2O_2$ and 0.2 μg mL$^{-1}$ HRP-Cu$^{2+}$ without (I) or with (II,III) reaction mixture (20 μL) of 1.0 μg mL$^{-1}$ PCSK9-DNA probes, 0.1 μM block-primer, 0.3 μM padlock, 0.625 U μL$^{-1}$ splint R ligase, 0.025 U μL$^{-1}$ phi 29 polymerase, 1.0 mM BSA and 2.0 mM dNTPs in absence (II) and presence (III) of 0.5 μg mL$^{-1}$ PCSK9-Ab. Error bars were estimated from three parallel experiments.

enzymatic switch could be employed to detect the molecules that could trigger RCA process.

To demonstrate the feasibility of the enzymatic switch for detection of RCA-related molecules, the primer of RCA was firstly used as an analyte (Fig. 3d). As expected, after the RCA product was incubated with the mixture of HRP-Cu$^{2+}$ and luminol-$H_2O_2$, the chemiluminescent intensity increased with the increasing primer concentration (Fig. 3e), and the plot of $\Delta I$ versus the logarithm of primer concentration showed good linearity in a range of 100 fM–1 μM (Fig. 3f). The limit of detection (LOD) corresponding to three times standard deviation was calculated to be 20 fM, which was dozens or hundreds of times lower than those from fluorescent assays with catalytic hairpin assembly and RCA amplification (1 pM)[43], and colorimetric assay with RCA printing technique (10 pM)[44], and it was even comparable with that from fluorescent assay using branched RCA strategy (10 fM)[45].

**Chemiluminescent response to PCSK9-Ab via P-RCAES.** As a proof-of-concept, a pair of PCSK9-DNA probes were used for specific identification of PCSK9-Ab (Fig. 3g). The PCSK9-DNA probes were prepared by covalently binding thiolated-DNA on PCSK9 protein via MBS linker[27], which was confirmed by PAGE analysis and protein-staining method (Supplementary Fig. 7). The amounts of PCSK9-DNA probes were optimized to be 2 ng mL$^{-1}$, at which the concentrations of probes were 10 times higher than PCSK9-Ab, and the chemiluminescent intensity was close to the maximum value (Supplementary Fig. 8). With the P-RCAES, the chemiluminescent intensity increased with the increasing concentration of PCSK9-Ab (Fig. 3h), which showed a good linear relationship with the logarithm of PCSK9-Ab concentration ranging from 1 pg mL$^{-1}$ to 1 μg mL$^{-1}$ (Fig. 3i). When the concentration of PCSK9-Ab was higher than 1 μg mL$^{-1}$, the chemiluminescent intensity would decrease due to the Hook effect which was common in homogeneous system[46]. The LOD of

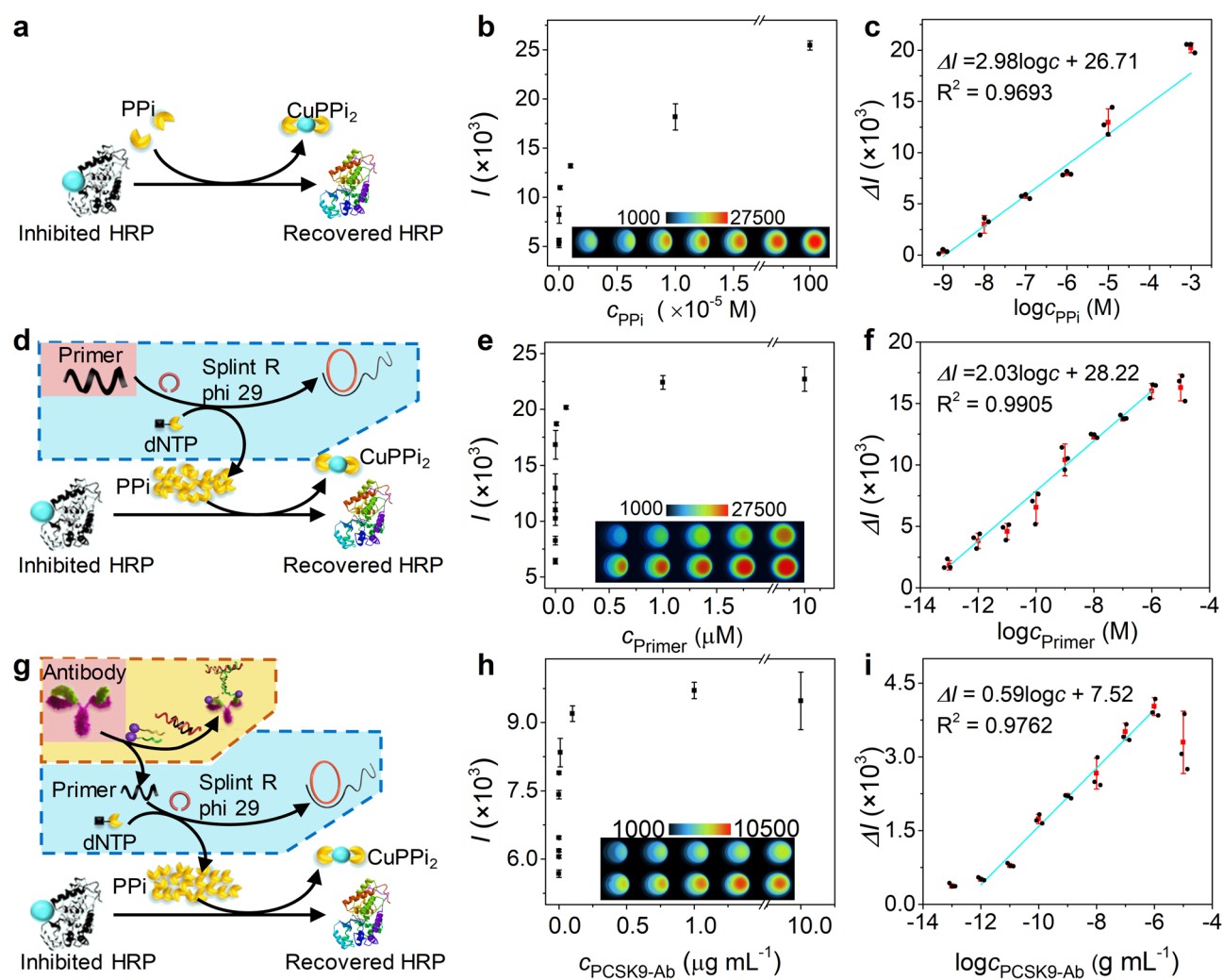

**Fig. 3 Chemiluminescent responses to pyrophosphate anions, primer and PCSK9-Ab. a** Schematic diagram of activation of enzymatic switch by pyrophosphate anions. **b** Chemiluminescent intensity ($I$) and **c** relative chemiluminescent signal ($\Delta I$) as function of logarithm of pyrophosphate anions concentration in 10 mM pH 7.2 Tris-HCl containing 200 ng mL$^{-1}$ HRP-Cu$^{2+}$, 0.2 mM luminol and 1 mM H$_2$O$_2$. **d** Schematic diagram of activation of enzymatic switch by a primer-induced rolling circle amplification. **e** Chemiluminescent intensity and **f** relative chemiluminescent signal as function of logarithm of primer concentration in 10 mM pH 7.2 Tris-HCl buffer (50 μL) containing 0.2 μg mL$^{-1}$ HRP-Cu$^{2+}$, 0.2 mM luminol, 1 mM H$_2$O$_2$ and the reaction mixtures (20 μL) of 0.3 μM padlock, 0.625 U μL$^{-1}$ splint R ligase, 0.025 U μL$^{-1}$ phi 29 polymerase, 1.0 mM BSA, 2.0 mM dNTPs with different concentrations of primer for 1-h incubation at 37 °C. **g** Schematic diagram of chemiluminescent assay of PCSK9-Ab via proximity-rolling circle activated enzymatic switch. **h** Chemiluminescent intensity and **i** relative chemiluminescent signal as function of logarithm of PCSK9-Ab concentration in 10 mM pH 7.2 Tris-HCl (50 μL) containing 0.2 μg mL$^{-1}$ HRP-Cu$^{2+}$, 0.2 mM luminol, 1 mM H$_2$O$_2$ and the reaction mixtures (20 μL) of 1.0 μg mL$^{-1}$ PCSK9-DNA probes, 0.1 μM block-primer, 0.3 μM padlock, 0.625 U μL$^{-1}$ splint R ligase, 0.025 U μL$^{-1}$ phi 29 polymerase, 1.0 mM BSA and 2.0 mM dNTPs with different concentrations of PCSK9-Ab for 1-h incubation at 37 °C. Error bars were estimated from three parallel experiments.

PCSK9-Ab was calculated to be 1 pg mL$^{-1}$ (equal to 18 fM) based on three times standard deviation, which was at least 3 orders of magnitude lower than those reported previously for antibody detection with various amplification strategies (Supplementary Table 2). The ultrahigh sensitivity of this proposed assay should be attributed to the RCA-assisted mass-production of PPi. In this work, the padlock template contained 44 bases. After RCA for over 700 cycles in 1 h[36], each antibody could produce more than 30000 PPi to activate the HRP. Besides, the specificity of the CLA was also evaluated by detecting solutions containing 1 μg mL$^{-1}$ of AFP-Ab, CEA-Ab, proBNP-Ab, and PCSK9-Ab, respectively. Only the sample containing PCSK9-Ab showed distinct signal (Supplementary Fig. 9), indicating good selectivity of the proposed CLA.

**CLA of PCSK9-Ab secreted from 6A6 hybridoma cells.** Hybridoma technique normally employs ELISA method to detect the secreted antibody in supernatant after specific hybridoma cells are cultured for 1–2 weeks. During this period the cells undergo clonal expansion and antibody secretion for meeting the ELISA-detectable concentration, which greatly limits the efficiency of hybridoma technique. The high sensitivity and specificity of the proposed CLA with P-RCAES provided a good substitute for ELISA to detect the secreted antibody in supernatant of single specific hybridoma cell without clonal expansion. Thus, we examined the assay performance of CLA for detecting PCSK9-Ab secreted from 6A6 hybridoma cells. The single cell was obtained by limiting dilution on 96-well plate, and the supernatants of 6A6 hybridoma cells with culture for different

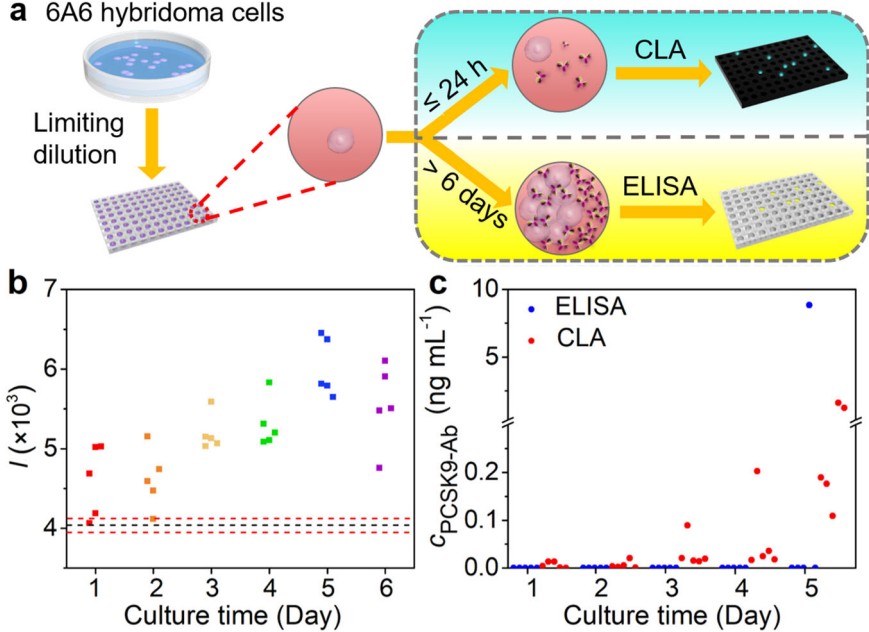

**Fig. 4 Assay of PCSK9-Ab secreted from 6A6 hybridoma cells. a** Schematic diagram of the detection of PCSK9-Ab secreted from 6A6 hybridoma cells by chemiluminescent assay and ELISA. **b** Chemiluminescent assay of PCSK9-Ab secreted from 6A6 hybridoma cells after culture for different days. **c** Concentration of secreted PCSK9-Ab after culturing 6A6 hybridoma cells for different days.

days were detected with both the CLA and ELISA (Fig. 4a). The PCSK9-Ab secreted from 6A6 hybridoma cells could not be detected by ELISA until 6 days after culture (Supplementary Fig. 10), while the proposed CLA could well detect the PCSK9-Ab from single 6A6 hybridoma cell after culture for only 1 day (Fig. 4b), providing a quick technique for screening of specific hybridoma cells. Compared to the sample cultured for 5 days, the supernatant from 6-day culture showed slightly lower chemiluminescent signals due to the Hook effect, which could be solved by using higher concentrations of PCSK9-DNA probes.

The amount of PCSK9-Ab secreted from 6A6 hybridoma cells could be calculated with the calibration curves of both ELISA (Supplementary Fig. 11) and CLA (Supplementary Fig. 12). 25 samples were randomly collected at different days after cell culture. Except one sample at day 1, PCSK9-Ab could be detected in all samples by CLA. On the contrary, only one sample collected at day 5 could be detected by ELISA (Fig. 4c and Supplementary Fig. 13), in which PCSK9-Ab concentration was 8.8 ng mL$^{-1}$. As mentioned above, the proposed CLA suffered from the Hook effect at high antibody concentration, thus the detection results for the samples collected at day 5 were lower than that by ELISA. However, the pg mL$^{-1}$ level of PCSK9-Ab detected by CLA at early stage of the culture should be credible, because the recovery test of CLA for PCSK9-Ab in supernatant of hybridoma cells showed a recovery range from 80.4% to 124.5% with the relative standard deviations (RSD) less than 14.9% (Supplementary Table 3). We observed the state of cells, and found that 1-day culture did not lead to the proliferation of the hybridoma cell in each well. This indicated that the proposed CLA was able to detect PCSK9-Ab secreted from single 6A6 hybridoma cell. According to the detection results, single 6A6 hybridoma cell could secret 0.2-2.7 pg of antibody in 24 h, which was consistent with the results reported previously[8], further demonstrating the reliability of the proposed CLA with P-RCAES. In view of the high sensitivity, we tried to use the CLA for further examining the amount of PCSK9-Ab secreted from single 6A6 hybridoma cell with 3-h, 6-h, and 12-h culture. Only part of the samples collected after 12-h culture showed the presence of PCSK9-Ab (Supplementary Fig. 14).

The reliability of CLA was further evaluated by detecting PCSK9-Ab in the culture medium of cells secreting PCSK9-Ab (6A6 hybridoma cells), or the culture medium of cells not secreting PCSK9-Ab, such as Hela cells, 3H2 hybridoma cells, and nonsecretor hybridoma cells, respectively. Only the supernatant of 6A6 hybridoma cells exhibited an obvious chemiluminescent signal, while the supernatant of non-PCSK9-Ab-secreting cells did not show obvious signal in comparison with the blank medium without cells (Supplementary Fig. 15). This result indicated the good reliability of CLA for detecting PCSK9-Ab as well as its application in screening of specific hybridoma cells.

**Chemiluminescent screening of specific hybridoma cells at single cell level.** 32 single hybridoma cell samples distributed in individual wells were tested by CLA at 1-day culture, ELISA at 6-day culture and microscopy to verify the performance of CLA for screening the specific hybridoma cells (Fig. 5 and Supplementary Figs. 16 and 17). At 1-day culture 22 samples showed PCSK9-Ab positivity, among which 20 samples were positive at 6-day culture, meanwhile 3 negative samples at 1-day culture could be detected to be positive at 6-day culture (Fig. 5). The positive matching rate of CLA at 1-day culture and ELISA at 6-day culture was 91%. After 1-day culture, all the single hybridoma cells kept in good condition without cell proliferation, but only 23 samples showed cell proliferation at 6-day culture (Supplementary Fig. 16). Considering the apoptosis of cells in sample 1 during 6-day culture, the matching rate reached 95%, indicating the good accuracy of CLA for specific hybridoma cell screening at single cell level. The false-negative results for samples 2, 3, and 5 at 1-day culture and sample 14 at 6-day culture could be attributed to the lower concentration of secreted PCSK9-Ab than the limits of detection (Supplementary Fig. 17). Although samples 6 and 15 showed good cell clonal expansion, no PCSK9-Ab was detected in their supernatant by both CLA and ELISA, indicating that they were PCSK9-Ab-nonsecreting hybridoma cells. Interestingly, hybridoma cells possessed great difference in the ability of secretion and expansion. For example, samples 8 and

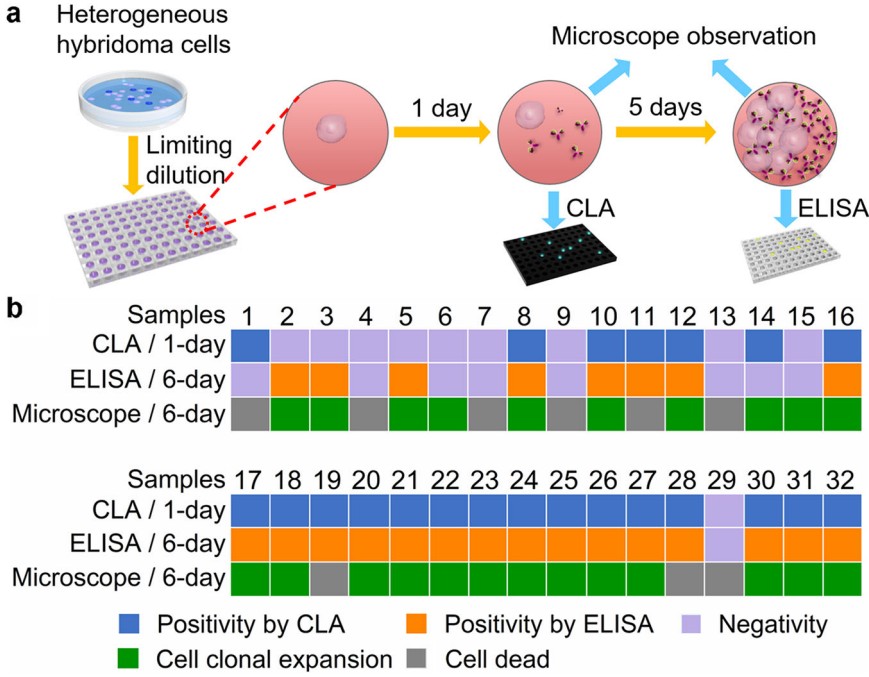

**Fig. 5 Screening of specific hybridoma cells. a** Schematic diagram of the chemiluminescent screening of specific hybridoma cells by chemiluminescent assay and ELISA. **b** Detection of PCSK9-Ab in supernatants of single hybridoma cell samples at 1-day culture by chemiluminescent assay and 6-day culture by ELISA, and the cell state at 6-day culture.

32 showed positive results from both CLA and ELISA, but the clonal ability of sample 8 was obviously stronger than sample 32.

In conclusion, a proximity-rolling circle activated enzymatic switch has been designed for ultrasensitive chemiluminescent detection of antibody and quick screening of specific hybridoma cells at single-cell level. The P-RCAES can be operated in homogeneous system by the series association of target antibody-induced proximity hybridization, RCA primer release, primer-triggered RCA, RCA-assisted mass-production of PPi, and enzyme activation by PPi. This cascade reactions can produce more than 30,000 PPi for each target antibody in 1-h incubation, leading to a LOD of PCSK9-Ab down to 18 fM. Moreover, owing to the ultrahigh sensitivity of this enzymatic switch, the screening of specific hybridoma cells can be completed within 1 day on the conventional 96-well plate, which greatly shortens the period of the production of high-quality monoclonal antibodies with hybridoma technology. The P-RCAES based chemiluminescent screening possesses attractive advantages of economy, simplicity, acceptable accuracy, and good expansibility without need of sophisticated instruments. This work provides opportunities for hybridoma cell screening and the development of cellular secretion detection, and can promote the production of immunotherapeutic agents against various diseases.

## Methods

**Materials**. HRP, luminol, $Cu(NO_3)_2$, $KNO_3$, $CaCl_2$, NaCl, $Mg(NO_3)_2$, $AlCl_3$, $Zn(NO_3)_2$, $AgNO_3$, $CoCl_2$, $Pb(NO_3)_2$, $Ni(NO_3)_2$, $FeCl_3$, $K_2HPO_4$, $KH_2PO_4$, PPi, tris(hydroxymethyl) aminomethane (Tris), tris(2-carboxyethyl) phosphine hydrochloride (TCEP), 3-maleimidobenzoic N-hydroxysuccinimide ester (MBS), polyethylene glycol 4000 (PEG 4000) and gel extraction kit were purchased from Sigma-Aldrich Co. (Shanghai, China). Human PCSK9 protein as well as HRP-labelled goat anti-mouse IgG was obtained from Abcam Co. (Shanghai, China). PCSK9-Ab was obtained from FANTIBODY (Chongqing, China). PCSK9 peptide 1-KLH: EGRVMVTDFENVPEEDGTRFHRQAS was obtained from Jier Biochem Co. (Shanghai, China). Splint R ligase, phi 29 DNA polymerase, dNTPs, ATP and corresponding buffer were purchased from New England BioLabs (Beverly, MA, U.S.A.). Foetal bovine serum (FBS), RPMI-1640 medium, DMEM medium, BCA protein kit, fast silver stain kit, and TMB Kit were bought from KeyGEN BioTECH Corp., Ltd (Jiangsu, China). HAT supplement (100×), HEPES buffer, sodium pyruvate, L-glutamine and PS (10 U $mL^{-1}$ penicillin and 10 mg $mL^{-1}$ streptomycin) were purchased from ThermoFisher (Shanghai, China). Hydrogen peroxide (30%), bovine serum albumin (BSA), and all DNA oligonucleotides were obtained from Sangon Biotechnology Co., Ltd. (Shanghai, China). The DNA sequences were listed in Supplementary Table 1. The reaction buffer (pH 7.2) contained 500 mM Tris-HCl, 100 mM KCl, 100 mM $(NH_4)_2SO_4$, 100 mM ATP and 40 mM DTT. Ultrapure water from a Millipore water purification system (Milli-Q, Millipore) was used for all experiments.

**Apparatus**. BioSpectrum 615 Imaging System with a cooled low-light CCD camera was used for chemiluminescent imaging to collect the mean pixel intensity within a circle. Chemiluminescence spectra was measured on F-7000 spectrometer (HITACHI, Japan). Absorption spectra was recorded on UV-3600 UV−vis-NIR spectrophotometer (Shimadzu Company, Japan). Chemiluminescent kinetic curves were obtained on MPI-A multifunctional electrochemical and chemiluminescent analytical system with PMT 600 (Xi'an Remex Analytical Instrument Co., Ltd. China). The gel electrophoresis was performed on Mini-PROTEAN Tetra System (Bio-RAD, USA) and imaged on Biorad ChemDoc XRS (Bio-Rad, USA). ELISA results were read out by Multiskan Sky (Thermo Fisher, USA).

**Preparation of HRP-$Cu^{2+}$ complex**. After HRP (10 μg $mL^{-1}$) was incubated with $Cu^{2+}$ (6 mM) at 37 ºC for 30 min, excess $Cu^{2+}$ was removed by ultrafiltration (50 kDa). The concentration of HRP-$Cu^{2+}$ complex was measured with BCA protein kit, which was then diluted to 1 μg $mL^{-1}$ and stored at 4 ºC for further use.

**Preparation of PCSK9-DNA probes**. Briefly, PCSK9 (40 μL, 1 mg $mL^{-1}$) was firstly reacted with a 40-fold excess of MBS in PBS (10 mM, pH 7.2) with a total volume of 100 μL for 2 h at room temperature to obtain PCSK9-MBS by ultrafiltration. Meanwhile, 150-fold molar excess of TCEP was used to reduce thiolated DNA (DNA 1 or DNA 2, 10 μL, 100 μM) in PBS (10 mM, pH 5.5) with a total volume of 200 μL for 2 h at room temperature. After removing extra TCEP by ultrafiltration, the reduced DNA 1 or DNA 2 was mixed with PCSK9-MBS obtained-above to incubate for 2 h at room temperature, and the PCSK9-DNA probes were purified by ultrafiltration. The concentrations of PCSK9-DNA probes were calibrated with the BCA protein assay kit. The successful preparation of PCSK9-DNA probes was confirmed by native polyacrylamide gel electrophoresis (PAGE) analysis and protein-staining method.

**Preparation of block-primer**. After primer and block were mixed at a ratio of 1:2 to react at 95 ºC for 5 min, block-primer was obtained, separated with gel extraction kit, and stored at 4 ºC.

**Preparation of hybridoma cells**. Spleen cells, which were obtained from PCSK9 peptide1-KLH immunized BALB/c female mice, SP2/0 myeloma cells and BALB/c mice peritoneal macrophages (feeder cells) were supplied by FANTIBODY (Chongqing, China). After spleen cells were fused with SP2/0 myeloma cells by mixing them at a ratio of 1:5 and adding dropwise 0.8 mL PEG 4000 solution (1 g mL$^{-1}$) within 1 min to incubate at 37 °C for 1.5 min, 29.2 mL of preheated serum-free RPMI-1640 was added into the mixture incubate at 37 °C for 5 min. The obtained hybrid cells were dispersed in 50 mL DMEM with 20% FBS, and mixed with feeder cells and then HAT supplement. After 1-week culture at 37 °C, heterogeneous hybridoma cells were obtained, which contained specific antibody-secreting cells, nonspecific antibody-secreting cells or nonsecretors. In order to establish the chemiluminescent screening procedure, the obtained heterogeneous hybridoma cells were conducted by subclone process with ELISA screening until they exhibited positive result, which were named as 6A6 hybridoma cells.

**Electrophoresis analysis**. 8% native polyacrylamide gel was prepared using 1 × TBE buffer to load the mixtures, which were prepared by mixing 5 μL samples with 1.5 μL 5× loading buffer and 1 μL SYBR$^{TM}$Gold dye to stand for 3 min. The gel electrophoresis was run at 100 V for 60 min in 1 × TBE buffer and visualized by a Biorad ChemDoc XRS (Bio-Rad, USA).

**Chemiluminescent response to primer**. After the mixtures containing 10 μL primer at different concentrations and 1 μL 6 μM padlock were heated at 95 °C for 5 min and cooled to room temperature, 9 μL solution containing splint R ligase (25 U μL$^{-1}$, 0.5 μL), phi29 polymerase (10 U μL$^{-1}$, 0.5 μL), reaction buffer (1 μL), dNTPs (10 mM for each of dATP, dGTP, dCTP, and dTTP, 1 μL), BSA (20 mM, 1 μL) was added in the mixture to incubate at 37 °C for 1 h for performing the RCA reaction, which was terminated at 65 °C for 10 min. Afterward, 10 μL HRP-Cu$^{2+}$ (1 μg mL$^{-1}$) was added into the mixture for 5-min incubation to perform the chemiluminescence measurement with an exposure time of 1 min by addition of 20 μL freshly prepared mixture of H$_2$O$_2$ (2.5 mM) and luminol (0.5 mM).

**Chemiluminescent response to PCSK9-Ab**. 10 μL PCSK9-Ab with different concentrations were firstly added into 10 μL mixture of PCSK9-DNA probes (10 μg mL$^{-1}$, 2 μL), block-primer (2 μM, 1 μL), padlock DNA (6 μM, 1 μL), dNTPs (10 mM for each of dATP, dGTP, dCTP, and dTTP, 1 μL), reaction buffer (1 μL), BSA (20 mM, 1 μL), splint R ligase (25 U μL$^{-1}$, 0.5 μL), phi 29 polymerase (10 U μL$^{-1}$, 0.5 μL), and 2 μL water to incubate for 1 h at 37 °C for releasing the primer and performing the RCA reaction. After the reaction was terminated at 65 °C for 10 min, 10 μL HRP-Cu$^{2+}$ (1 μg mL$^{-1}$) was added to react for 5 min, and then 20 μL freshly prepared mixture of H$_2$O$_2$ (2.5 mM) and luminol (0.5 mM) was added to collect the chemiluminescent images by CCD with an exposure time of 1 min.

**Calibration curve for CLA of PCSK9-Ab**. 10 μL DMEM culture media with different concentrations of PCSK9-Ab were added into 10 μL mixture of PCSK9-DNA probes (0.1 μg mL$^{-1}$, 2 μL), block-primer (2 μM, 1 μL), padlock DNA (6 μM, 1 μL), dNTPs (10 mM for each of dATP, dGTP, dCTP, and dTTP, 1 μL), reaction buffer (1 μL), BSA (20 mM, 1 μL), splint R ligase (25 U μL$^{-1}$, 0.5 μL), phi 29 polymerase (10 U μL$^{-1}$, 0.5 μL) and 2 μL water to incubate at 37 °C for 1 h. After the RCA reaction was terminated at 65 °C for 10 min, 10 μL HRP-Cu$^{2+}$ (1 μg mL$^{-1}$) was added for 5-min reaction, and then 20 μL freshly prepared mixture of H$_2$O$_2$ (2.5 mM) and luminol (0.5 mM) was added to collect the chemiluminescent images by CCD with an exposure time of 1 min.

**Calibration curve for ELISA of PCSK9-Ab**. PCSK9 coated 96-well plate was firstly prepared by incubating the plate with 2 μg mL$^{-1}$ PCSK9 for 6 h at 37 °C. After a vigorous washing, the plate was immersed in 10 mg mL$^{-1}$ BSA for 2 h at 37 °C to block the unreacted sites. After the plate was washed, 50 μL PCSK9-Ab at different concentrations were added into wells to incubate at 37 °C for 2 h. After washing and drying the plate, 50 μL IgG-HRP (2 μg mL$^{-1}$) was delivered into the wells to incubate for 2 h at 37 °C for performing ELISA with TMB kit at 450 nm on Multiskan Sky (Thermo Fisher, USA).

**Assay of PCSK9-Ab secreted from 6A6 hybridoma cells**. After 6A6 hybridoma cells were distributed on six 96-well plates to allow single cell in a well by limiting dilution with culture medium, they were cultured with DMEM medium (200 μL) supplemented with 20% FBS, HEPES buffer (10 mM), sodium pyruvate (1 mM), L-glutamine (2 mM) and PS in an atmosphere of 5% CO$_2$ and 95% humidified air at 37 °C for 1, 2, 3, 4, 5 and 6 days, respectively. Then, the supernatant in each well was collected to detect the amount of PCSK9-Ab secreted from 6A6 hybridoma cells by CLA and ELISA.

Hela cells (KeyGEN Biotech, Nanjing, China), 3H2 hybridoma cells (secreting aminoterminal pro-brain natriuretic peptides monoclonal antibody, FANTIBODY, Chongqing, China), and nonsecretor hybridoma cells (FANTIBODY, Chongqing, China) were used as non-PCSK9-Ab-secreting cells for control experiment. The culture and CLA processes of these non-PCSK9-Ab-secreting cells were the same as that of 6A6 hybridoma cells.

**Chemiluminescent screening of specific hybridoma cells**. After heterogeneous hybridoma cells were distributed into the wells of two plates by limiting dilution, DMEM medium supplemented (200 μL) with 20% FBS, HEPES buffer (10 mM), sodium pyruvate (1 mM), L-glutamine (2 mM) and PS was added in each well to incubate in an atmosphere of 5% CO$_2$ and 95% humidified air at 37 °C for 6 days. 10 μL supernatant was then collected from each well at Day-1 to perform chemiluminescent screening of specific hybridoma cells at single cell level, while 50 μL supernatant was collected at Day-6 to perform ELISA screening.

**Statistics and reproducibility**. All quantitative data in the article and Supplementary Information were independently repeated three times and displayed as the mean ± SD. All the calculations were done in Origin or Excel.

**Reporting summary**. Further information on research design is available in the Nature Research Reporting Summary linked to this article.

## Data availability

Source data for all the graphs and charts in the main figures is provided as Supplementary Data 1 and any remaining information can be obtained from the corresponding author upon reasonable request.

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

## Acknowledgements

We gratefully thank the National Natural Science Foundation of China (21827812, 21635005, 21890741), Independent Research Foundation from State Key Laboratory of Analytical Chemistry for Life Science (5431ZZXM2006), and the Fundamental Research Funds for the Central Universities (14380209) for funding supply.

## Author contributions

H.X.J. and J.W. initiated the project. H.A., J.W., and H.X.J. conceived and designed the experiments. H.A., W.W.C., and W.C.X. performed the experiments. H.A. and J.W. analyzed the data. H.A., J.W., and H.X.J. co-wrote the paper. All authors discussed the results and commented on the manuscript.

## Competing interests

The authors declare no competing interests.
