## [Peer Review File · Communications Biology]

Reviewers' comments:

Reviewer #1 (Remarks to the Author):

Huangxian Ju et al. present and demonstrate a method for screening hybridoma cells via detecting an antibody (PCSK9-Ab) specifically secreted by these cells. For the detection part, they use two DNA probes that bind those antibodies and trigger linear rolling circle amplification (RCA). The trigger mechanism is incorporated into a so-called block primer that is otherwise protected from a circular DNA template. Once bound to the antibody, the two DNA probes act on the block primer (turn dsDNA into ssDNA via strand displacement) and activate RCA leading to production of magnesium pyrophosphates. That inorganic molecule is a well known by-product of any DNA amplification technique, and here it is used to chelate divalent copper ions that otherwise inhibit horse-radish peroxidase enzyme (HRP). In the presence of the antibodies, RCA leads to formation of the pyrophosphates, and so the HRP enzyme is active and is used to convert a substrate that provides a detection signal in a form of electro chemiluminescence (ECL). The authors first demonstrated the analytical approach that provided a limit of detection of 20 fM (1 hr RCA). They furthermore demonstrate that the antibody secreted from hybridoma cells could not be detected by ELISA until 6 days after culture while the proposed assay could well detect the PCSK9-Ab from "single" hybridoma cell after culture for only 1 day.

Although the limit of detection is appealing, the concept itself is cumbersome. From the sketch, it looks like the two DNA probes that bind the antibody at the very beginning can do the job (activate RCA) without the antibody. Thus I still cannot figure out the need for proximity, so this part should be explained better. As any amplification technique, RCA can be prone of spurious amplification, thus more stressing on the negative controls is required (I could not find a clearly specified negative control, which should be free of the Hybridoma cells as well as antibodies). I like the characterization of the HRP enzyme and it is novel to use the pyrophosphates to "steal" enzyme co-factor, well in this case the co-factor is inhibiting the enzyme. But wouldn't it be easier to implement branched RCA detecting total concentration of magnesium pyrophosphates or total DNA fluorescence?

Compared to ELISA, the proposed assay is faster (takes one day waiting instead of six) but it requires more hours of handling than simple ELISA unless all reagents (DNA probes, block primer, circular template, ligase, polymerase, HRP enzyme) can be placed at the same time in one pot? Finally yet importantly, the overall concept is based on findings of Mats Nilsson and is another application of RCA technique with an extension such as Chemiluminescent detection of HRP. This manuscript of course deserves a publication but in a more suitable journal such as *Analyst*, *Talanta*, *Biosensors & Bioelectronics* or similar.

Reviewer #2 (Remarks to the Author):

The manuscript on 'Chemiluminescent Screening of Specific Hybridoma Cells via a Proximity-Rolling Circle Activated Enzymatic Switch' addresses the prevalent and practical issue of massive screening required in antibody selection for diverse applications, especially considering the time and technical complexity (lengthy protocols and sophisticated infrastructure/instrumentation) involved. To address this gap, the authors have reported a time-saving 1-day chemiluminescent screening using the proposed P-RCAES strategy based on proximity rolling circle amplification method coupled to an enzymatic molecular switching.

Although hybridoma technology is broadly used for antibody production, it is not the exclusive method available for the same. It might be interesting at least to add some basic notes on the available alternate methods of antibody production (eg, recombinant technology, cell free synthesis, etc.) in the introduction section.

The authors bring in the notes on methods pertaining to homogenous analysis (integrated approach for assay capture and detection, without any separation / immobilization / washing), in the introduction. However, it needs to be clarified towards the end of the introduction whether the reported method in this study based on molecular switching falls under the category of homogeneous analysis or not (though this information is highlighted in Supplementary Table 2), for the clarity of broader audience.

In the line 58, the rolling of template in RCA has been mentioned as 'hundreds of times' – this generally depends (could sometimes be quite short or very long) on the parameters of time of rolling and available concentration of the reagents in the RCA mix.

In the introduction section (somewhere between lines 81 and 90), the authors need to also explain (or at least cite a suitable reference) the use of both splintR ligase as well as phi29 DNA polymerase, as the role of the other molecular components have been mentioned.

In the results section, the differences in the initial, inhibited and recovered chemiluminescent intensities have been appropriately provided as a baseline for readers new to the area. The reason for the choice of Cu²⁺ over Cu⁺ in the design of molecular (enzymatic) switch has also been justified. However, it would be interesting for non-analytical readers to get an idea about the details of background subtraction done for CLA (eg, as in page-9, Line 149) in deriving the numbers with the chemiluminescent intensities for Fig. 3 to 6 (at least in the Methods part or Supplementary Information), as these form the backbone for the detection platform reported.

All relevant optimization parameters have been brought in the supplementary information in a convincing manner.

The language and use of certain phrases need to be checked and appropriately revised to suit the broad range of readers. Some examples include:

Line 18, page-1: 'provides a candidate' – please alter with technical terminology here in the abstract.

Line 29, page-2: 'Although commonly used ELISA' instead of 'Although common used'

Line 368-369, page 20: '1 day' and '6 days' to be replaced by 'Day-1' and 'Day-6', respectively, in the Methods section.

I recommend this article for publication with minor revisions as suggested above.

Response to comments of reviewers

To Reviewer 1:

General comment: *Huangxian Ju et al. present and demonstrate a method for screening hybridoma cells via detecting an antibody (PCSK9-Ab) specifically secreted by these cells. For the detection part, they use two DNA probes that bind those antibodies and trigger linear rolling circle amplification (RCA). The trigger mechanism is incorporated into a so-called block primer that is otherwise protected from a circular DNA template. Once bound to the antibody, the two DNA probes act on the block primer (turn dsDNA into ssDNA via strand displacement) and activate RCA leading to production of magnesium pyrophosphates. That inorganic molecule is a well known by-product of any DNA amplification technique, and here it is used to chelate divalent copper ions that otherwise inhibit horse-radish peroxidase enzyme (HRP). In the presence of the antibodies, RCA leads to formation of the pyrophosphates, and so the HRP enzyme is active and is used to convert a substrate that provides a detection signal in a form of electro chemiluminescence (ECL). The authors first demonstrated the analytical approach that provided a limit of detection of 20 fM (1 hr RCA). They furthermore demonstrate that the antibody secreted from hybridoma cells could not be detected by ELISA until 6 days after culture while the proposed assay could well detect the PCSK9-Ab from “single” hybridoma cell after culture for only 1 day.*

Response: Thank you for the comments. This work demonstrates an effective chemiluminescent (CL) screening protocol for specific hybridoma cells on conventional 96-well plate via detecting the specific antibody secreted from these cells. Benefiting from the low detection limit of the antibody analytical approach, this protocol can shorten the screening procedure of hybridoma cells to 1 day without tedious cloning process.

Comment 1: *Although the limit of detection is appealing, the concept itself is cumbersome. From the sketch, it looks like the two DNA probes that bind the antibody at the very beginning can do the job (activate RCA) without the antibody. Thus I still cannot figure out the need for proximity, so this part should be explained better. As any amplification technique, RCA can be prone of spurious amplification, thus more stressing on the negative controls is required (I could not find a clearly specified negative control, which should be free of the Hybridoma cells as well as antibodies).*

Response: In this work, two DNA sequences containing 6 complementary bases (purplish

red regions in Supplementary Table 1) are used to label the antigen, as one pair of probes to recognize the binding sites of one antibody, which leads to proximity hybridization of DNA 1 and DNA 2 to form a proximity-ligated oligonucleotide sequence (refs 17-32). The formed sequence can then hybridize with block to release the primer from block-primer. This is a key step of the proposed switch. In the absence of target antibody, it is impossible to form stable duplex between such short complementary sequences in DNA 1 and DNA 2. For better understanding, a new Supplementary Figure 1 has been supplied to explain the antibody-induced proximity hybridization, and the related description has also been modified in page 5 line 88. Correspondingly, the original Supplementary Figures 1-13 have been adjusted to Supplementary Figures 2-14.

Supplementary Figure 1. Illustration of the antibody-induced proximity hybridization.

As for the negative control of RCA, the result shown in Fig. 2d, column II can be considered as the negative control of RCA or P-RCAES for PCSK9-Ab detection (page 7 line 128 to page 8 line 130). In addition, the control result was also exhibited in the specificity study of CLA (page 11 lines 188-189 and Supplementary Figure 9).

According to this comment, we further used three non-PCSK9-Ab-secreting cells including Hela cells, 3H2 hybridoma cells (secreting aminoterminal pro-brain natriuretic peptides monoclonal antibody), and nonsecretor hybridoma cells as negative controls to evaluate the reliability of CLA. As shown in a new Supplementary Figure 15, the supernatant of 6A6 hybridoma cells (PCSK9-Ab-secreting cells) exhibited an obvious CL signal, while the supernatants of non-PCSK9-Ab-secreting cells did not show significant signal in comparison with the blank medium without cells, indicating good reliability of CLA for detecting PCSK9-Ab as well as its application in screening of specific hybridoma cells. The corresponding discussion has been added in page 14 lines 245-251, and page 21 lines 385-389 in the revised manuscript.

Supplementary Figure 15. CLA of PCSK9-Ab in culture medium of cells secreting or not secreting PCSK9-Ab with culture for 3 days. (1) blank medium without cells, (2-5) culture medium of Hela cells (2), 3H2 hybridoma cells (3), nonsecretor hybridoma cells (4) and 6A6 hybridoma cells (5).

Comment 2: *I like the characterization of the HRP enzyme and it is novel to use the pyrophosphates to “steal” enzyme co-factor, well in this case the co-factor is inhibiting the enzyme. But wouldn’t it be easier to implement branched RCA detecting total concentration of magnesium pyrophosphates or total DNA fluorescence?*

Response: This work aims at developing an ultrasensitive CL protocol for antibody detection and quick screening of specific hybridoma cells. As HRP catalyzed luminol-H₂O₂ reaction is the most developed and the commonly used CL system, this work uses HRP to design HRP-enzymatic switch (Fig. 2a) and the P-RCAES strategy to produce CL signal for the detection of secreted antibody (Fig. 1b).

In addition, pyrophosphate anion (PPi) is the by-product of dNTP transformation, thus the amount of PPi is much larger than that of DNA copies in a same RCA process (here 30000 PPi vs 700 DNA copies) (page 10 lines 186-187), hence the proposed CL assay with P-RCAES shows much lower LOD of PCSK9-Ab than other methods such as DNA based fluorescent assays (page 10 lines 181-184, and Supplementary Table 2).

What’s more, the PPi-activated enzymatic switch was carried out only by adding HRP-Cu²⁺ to RCA solution without separation and washing steps (See the “CL response to PCSK9-Ab” in Methods, page 20 lines 360-362), so it was as easy as the direct detection of magnesium pyrophosphates or DNA fluorescence.

Comment 3: *Compared to ELISA, the proposed assay is faster (takes one day waiting instead of six) but it requires more hours of handling than simple ELISA unless all reagents*

(DNA probes, block primer, circular template, ligase, polymerase, HRP enzyme) can be placed at the same time in one pot?

Response: The proposed CLA of PCSK9-Ab included 3 steps: (1) antibody-induced RCA, which was operated by adding target solution to the mixture solution of PCSK9-DNA probes, block-primer, padlock DNA, dNTPs, ligase and polymerase with an incubation of 1 h; (2) PPi-activated enzymatic switch, which was operated by adding HRP-Cu²⁺ to above RCA solution with an incubation of 5 min; (3) CL imaging assay, which was operated by adding the mixture of H₂O₂ and luminol to the reaction solution for collecting the CL images by CCD with an exposure time of 1 min (See the “CL response to PCSK9-Ab” in Methods, page 20 lines 355-362). Therefore, this assay could be performed in three sequential solution additions within 66 min without washing and separation handlings, which were required in ELISA, and the proposed CLA method was simple with shorter assay time (page 14 line 253 to page 15 line 258) than ELISA.

Comment 4: *Finally yet importantly, the overall concept is based on findings of Mats Nilsson and is another application of RCA technique with an extension such as Chemiluminescent detection of HRP. This manuscript of course deserves a publication but in a more suitable journal such as Analyst, Talanta, Biosensors & Bioelectronics or similar.*

Response: Indeed, this work describes a “1-day chemiluminescent screening protocol” for screening of specific hybridoma cells at single cell level on conventional 96-well plate via the newly designed proximity-rolling circle activated enzymatic switch (P-RCAES) strategy (Fig. 1), as the title and pointed out by reviewer 2. The P-RCAES is operated by series association of target antibody-induced proximity hybridization, DNA hybridization-triggered RCA, RCA-assisted mass-production of PPi, PPi-activated enzymatic switch, and enzymatic CL generation (detail description in page 5 line 85 to page 6 line 96). Although the proposed P-RCAES employed RCA technique to produce PPi, the essence of P-RCAES was not RCA but the activation reaction of HRP by PPi. Actually, RCA could here be replaced with any other DNA polymerization reactions to perform the amplification, because PPi was produced during the dNTP transformation.

Compared with ELISA screening, the proposed CLA protocol via P-RCAES greatly shortened the screening period of specific hybridoma cells along with attractive advantages of economy, simplicity, acceptable accuracy and good expansibility without need of sophisticated instruments. Thus, this work may provide new opportunities for hybridoma cell screening and development of cellular secretion detection. Therefore, we believe the

manuscript will be of interest to the broad audience of Communications Biology and be suitable to publish in Communications Biology

To Reviewer 2:

General comment: *The manuscript on ‘Chemiluminescent Screening of Specific Hybridoma Cells via a Proximity-Rolling Circle Activated Enzymatic Switch’ addresses the prevalent and practical issue of massive screening required in antibody selection for diverse applications, especially considering the time and technical complexity (lengthy protocols and sophisticated infrastructure/instrumentation) involved. To address this gap, the authors have reported a time-saving 1-day chemiluminescent screening using the proposed P-RCAES strategy based on proximity rolling circle amplification method coupled to an enzymatic molecular switching.*

Response: Thank you for the comments. Yes, this work describes a “1-day chemiluminescent screening protocol” for screening of specific hybridoma cells at single cell level on conventional 96-well plate via P-RCAES.

Comment 1: *Although hybridoma technology is broadly used for antibody production, it is not the exclusive method available for the same. It might be interesting at least to add some basic notes on the available alternate methods of antibody production (eg, recombinant technology, cell free synthesis, etc.) in the introduction section.*

Response: According to the comment, we have added the related information “Monoclonal antibodies that have been proven to be the powerful therapeutic agents against various diseases can be produced by different methods such as hybridoma technology, phage display, ribosome display, mRNA display, and B-cell immortalization technology” in the beginning of Introduction section in page 3 lines 22-25.

Comment 2: *The authors bring in the notes on methods pertaining to homogenous analysis (integrated approach for assay capture and detection, without any separation / immobilization / washing), in the introduction. However, it needs to be clarified towards the end of the introduction whether the reported method in this study based on molecular switching falls under the category of homogeneous analysis or not (though this information is highlighted in Supplementary Table 2), for the clarity of broader audience.*

Response: Thank you for the suggestion. We have added the description of “homogeneous

CL assay” in Abstract (page 2 line 15) and “homogeneous antibody detection” in Introduction (page 4 line 59 and page 5 line 71) to clarify that the proposed assay was homogeneous analysis.

Comment 3: *In the line 58, the rolling of template in RCA has been mentioned as ‘hundreds of times’ – this generally depends (could sometimes be quite short or very long) on the parameters of time of rolling and available concentration of the reagents in the RCA mix.*

Response: According to the comment, the description has been revised to “up to hundreds of times” in page 4 line 62.

Comment 4: *In the introduction section (somewhere between lines 81 and 90), the authors need to also explain (or at least cite a suitable reference) the use of both splintR ligase as well as phi29 DNA polymerase, as the role of the other molecular components have been mentioned.*

Response: Here, splint R ligase could ligate the padlock to the ring template, and phi 29 polymerase could extend the primer along the ring template with dNTPs to produce long oligonucleotide amplicon (ref 34) as well as a huge mass of PPi. This description has been added in Results and discussion section in page 6 lines 91-93.

Comment 5: *In the results section, the differences in the initial, inhibited and recovered chemiluminescent intensities have been appropriately provided as a baseline for readers new to the area. The reason for the choice of Cu^{2+} over Cu^+ in the design of molecular (enzymatic) switch has also been justified. However, it would be interesting for non-analytical readers to get an idea about the details of background subtraction done for CLA (eg, as in page-9, Line 149) in deriving the numbers with the chemiluminescent intensities for Fig. 3 to 6 (at least in the Methods part or Supplementary Information), as these form the backbone for the detection platform reported.*

Response: The CL responses to PPi, primer and PCSK9-Ab, and the raw CL intensities and images corresponding to different concentrations were shown in Fig. 3b, 3e and 3h. Because the relative CL intensity change (ΔI , referring to the absolute signal subtracting the background) was entirely dependent on the target concentration, it was used to establish the linear detection curves (Fig. 3c, 3f and 3i). This description has been added in page 9 lines 154-156.

Comment 6: *The language and use of certain phrases need to be checked and*

appropriately revised to suit the broad range of readers. Some examples include:

Line 18, page-1: 'provides a candidate' – please alter with technical terminology here in the abstract.

Line 29, page-2: 'Although commonly used ELISA' instead of 'Although common used'

Line 368-369, page 20: '1 day' and '6 days' to be replaced by 'Day-1' and 'Day-6', respectively, in the Methods section.

Response: Thanks for the reminding. We have checked the writing, and revised “provides a candidate” to “provides a new screening method” in Abstract (page 2 line 19), “Although common used” to “Although commonly used” (page 3 line 32), and “1 day, 6 days” to “Day-1, Day-6” in Methods section (page 21 lines 394-395), respectively.

Other revised examples include: “building” to “the development of” in page 3 line 40, “methods has also been proposed for protein detection through” to “methods for protein detection was proposed through” in page 4 line 48 and “leading to a power tool for screening” to “thus it can be applied to screen” in page 5 line 72.

In addition, we have formatted the revised manuscript under the guidance of “revision file checklist”, including the conversion of all bar graphs to dot-plot format in Fig. 2cd and Fig. 3cfi, the addition of “Statistics and reproducibility” in Methods section (page 21 lines 397-399), the correction of title format, the addition of “Abstract” and “Introduction” headings, and the revision of unit format of dimensions as negative integers. The source data underlying the graphs presented in the main figures are also supplied in a separate excel file named “Supplementary Data”.

REVIEWERS' COMMENTS:

Reviewer #1 (Remarks to the Author):

I recommend to accept this manuscript.

Reviewer #2 (Remarks to the Author):

This revised manuscript can be considered for publication.